# Assessment of an Emergency Medicine System for Radiation Accidents in Korea: A State Survey of the Workers Involved the Medical Response to Radiation Accidents

**DOI:** 10.3390/ijerph18052458

**Published:** 2021-03-02

**Authors:** You Yeon Choi, Mihyun Yang, Younghyun Lee, Eunil Lee, Young Woo Jin, Ki Moon Seong

**Affiliations:** 1National Radiation Emergency Medical Center (NREMC), Korea Institute of Radiological & Medical Sciences (KIRAMS), Seoul 01812, Korea; c9640@kirams.re.kr (Y.Y.C.); mhy@kirams.re.kr (M.Y.); ylee@kirams.re.kr (Y.L.); ywjin@kirams.re.kr (Y.W.J.); 2Department of Environmental and Occupational Health, Graduate School of Public Health, Korea University, Seoul 02841, Korea; eunil@Korea.ac.kr; 3Department of Preventive Medicine, College of Medicine, Korea University, Seoul 02841, Korea

**Keywords:** emergency medicine system, radiation, licensee workers, inspectors, occupational role, regulatory system

## Abstract

Radiation emergency medicine systems are operated around the world to provide special care for the injured that require immediate medical attention in accidents. The objective of this survey was to evaluate people’s perception of those who design the emergency medical plan for radiation accidents and those who supervise it in Korea. A questionnaire survey was conducted on the people involved in a regulatory system for medical response in a radiation emergency. Of 150 survey recipients, 133 (88.7%) completed the survey, including 92 workers and 41 inspectors. The respondents expressed the view that the national emergency medical plan is prepared above the average level using a Likert-style scale of 1 to 5 (mean = 3.55, SD = 0.74). Interestingly, using the Mann–Whitney U test, it could be shown that inspectors evaluated the emergency medical system for radiation accidents more strictly in all of the questions than the licensee workers, especially on radiation medical emergency preparedness *(p* = 0.004) and the governmental regulatory policy for radiation safety (*p* = 0.007). For a more efficient system of radiation emergency medicine, licensee workers prioritized the workforce, whereas inspectors favored laws and regulations for safety. The survey results show different perspectives between inspectors and licensee workers, which stem from the actual properties of each occupational role in the regulatory system for radiation medical emergency. These data could be utilized for communication and interaction with relevant people to improve the medical response preparedness against radiation accidents.

## 1. Introduction

After the Fukushima nuclear power plant (NPP) accident in Japan in March 2011, the public was overwhelmed by severe anxiety about radiation exposure in Korea, which resulted in temporary closures of schools, massive selling of masks against radioactive dust, and a higher distrust of the health and safety of agricultural and marine products from Japan [1,2]. The people seriously considered the radiation safety issue due to the geographical proximity to the accident site and the tens of nuclear power plants and industrial applications in the Korean peninsula [3]. The people demanded the government prepare a consolidated regulatory system focused on radiation safety. The roles and responsibilities of the Nuclear Safety and Security Commission (NSSC), a governmental organization for nuclear safety and regulation in Korea similar to the United States Nuclear Regulatory Commission (U.S NRC), have been intensified to oversee affairs regarding the radiation emergency preparedness and response, including the medical emergency response system [4,5]. A medical response network was constructed with 24 hospitals for radiation emergencies in Korea. More than 600 medical staff elements and first responders have been educated and trained for various kinds of radiation emergencies. The role of local governments in Korea for the radiation emergency medicine system is to implement the protective actions based on providing assistance for the medical system, including the support of emergency service systems for patient transfer, contamination screening and decontamination of residents before and after moving into shelter, psychological counseling, and potassium iodine distribution and monitoring of side effects. As a previously reported study showed, the network of emergency medicine will work more efficiently for those injured in a crisis of radiation when the initial medical response is appropriate at the sites of accidents in radiation facilities [6].

As stated in the legal Article 45 of the Act on Measures for the Protection of Radiation Disasters in Korea, all nuclear licensees, such as companies and institutes having nuclear or radiation facilities, should establish medical preparedness plans for protection and human care in a radiation emergency, and implement drills at their facilities following the plan. This is in accordance with the international recommendation for safety requirements assigned by International Atomic Energy Agency (IAEA) as emergency preparedness categories I, II, III, and IV [7]. The emergency medicine plan for radiation facilities also needs to be examined and inspected from the design to the execution of emergency drills. To become more efficient against radiation accidents, the plan needs to be continually examined and supplemented. As entrusted by the NSSC since 2015, a revision and inspection of the radiation emergency medical plan has been legally performed by the specialists of the National Radiation Emergency Medical Center (NREMC), which provides specialized training courses for the first responders and medical staff in a radiation emergency, such as the Radiation Emergency Assistance Center/Training Site (REAC/TS) [8].

To strengthen the medical care for worker victims of occupational radiation exposure due to accidents, a unique regulatory system for radiation emergency medicine was legally introduced to all of the companies and institutes with nuclear/radiation facilities in Korea. They should prepare the emergency medicine plan for nuclear/radiological accidents, including the rescue of the injured, first aid, and transfer of patients to hospitals. The plan should be designed taking into account the scale, structure, and number of workers of each facility, according to references such as the IAEA guidance [9,10]. Specialists should review the medical response plan for radiation emergencies and regularly inspect the implementation of the plan at on-site facilities, accompanying every radiation emergency drill. Inspectors and licensee workers together have developed and complemented an emergency medicine system for radiation accidents through many assessments and discussions, and this communication is thought to be essential for the successful design of a plan. This is a unique regulatory system for medical responses to radiation emergencies that could not be found in other countries. Moreover, there is no internationally stipulated standard to ensure a prompt and efficient medical intervention for radiation accidents. Thus, this plan could be the basis for radiation safety regulation by other national authorities.

The companies and institutes with radiation facilities should prepare their own emergency medical plans, which should be examined by law. The NREMC has investigated the individual emergency plans for each facility and assessed whether they could be appropriately executed under a radiation emergency. The emergency medicine plans are not identical, and can be very complicated, depending on the scale of the facilities, the structure of the building, and the number of workers. This survey aimed to evaluate the perception of people involved in the execution and examination of the emergency plan of medical preparedness for a radiation emergency. This kind of survey study of the perception of societal issues is usefully applied to investigate the gap-filling of perspectives that are different between interested stakeholders, including the public [11,12]. Previous research about the social anxiety and poor quality of life after the NPP accident by Japanese researchers reflect concerns about exposure to radiation [13,14]. Communication with the perception data is considered as an effective way to solve the problem of knowledge deficiency towards formulating the public’s understanding.

## 2. Materials and Methods

### 2.1. Survey Methods

A questionnaire (Appendix A) was devised by the authors and validated for this study by a statistician, epidemiologist, and health physicist. It was pilot tested by a small group of people, including medical staff, health physicists, and administrators in emergency medical planning and then modified before execution as a large-scale survey. The questionnaire consisted of 18 questions, including 6 questions addressing the perception of the radiation emergency medicine system, 5 questions assessing the knowledge of radiation, and 7 questions addressing respondents’ personal information (i.e., sex, age, academic degree, affiliation, occupational role, career periods, and relevance to medical emergencies). A questionnaire survey was conducted on the people involved in the radiation medical emergency plan. Some are licensee workers preparing the emergency plan in each organization, such as companies and institutes with nuclear or radiation facilities, not hospitals in the radiation emergency network, and the others are specialists (inspectors) that examine and evaluate the plan. It was administered to the participants during a workshop for medical preparedness for radiation emergencies in December 2019. Some responses were collected at the workshop and the others were delivered by e-mail as a scanned file to increase the response rate for the survey. The responses were measured using a Likert-style scale of 1 to 5. The responses were anonymous and would not influence the performance in a specific study course. This study was approved by the institutional review board (IRB: KIRAMS 2020-04-009) of the Korea Institute of Radiological and Medical Sciences and conducted following the Declaration of Helsinki.

### 2.2. Statistical Analysis

Statistical analysis was performed using SPSS software (version 23; SPSS Inc., Chicago, IL, USA). Univariate and multivariate linear regression was performed to explore variables related to the answer to “Do you agree that the national plan of emergency medicine is well-prepared for nuclear or radiation accidents?” A Mann–Whitney U test was used to compare the question #8–#11 score of inspectors with licensee workers. Pearson’s chi-square analysis was used to analyze the difference in the proportion of medical emergency work between inspectors and licensee workers. A *p*-value < 0.05 was considered to be statistically significant.

## 3. Results

### 3.1. General Characteristics of Respondents

One-hundred thirty-three of the participants in the workshop for radiation emergency medicine replied to the survey; 150 questionnaire sheets were distributed (88.7% collection rate). Radiation emergency medicine (REM) staff in our survey of the workshop are representative workers entrusted by each company and institute with the license. In total, 689 persons have been enrolled as REM staff and affiliated with various companies and institutes, including a nuclear power plant, nuclear fuel manufacturing company, large-scale irradiation facility, radio waste disposal company, and R&D institute in Korea (as of 19.03.31). We believed that the survey results showed the representative opinion of the REM staff, considering the response ratio (133/689). Our study subjects consist of workers with various occupations with different affiliations. They are radiation safety officers, administrators, medical doctors, and medical researchers, including health physicists, radiation biologists, and medical specialists. Approximately 70% of the respondents were radiation workers affiliated with companies and institutes undergoing a revision and inspection of the emergency medicine plan, and 43% of them worked for a nuclear power company. Based on the score of five questions assessing the knowledge of radiation, they had the basic knowledge necessary for radiation safety, given that 90.4% answered more than three questions correctly. Over half of them had more than six years of experience with radiation emergency preparedness (Table 1).

### 3.2. Factors Associated with the Evaluation of the Status of Radiation Medical Emergency Preparedness

People related to medical preparedness of radiation emergencies were asked to evaluate the medical preparedness status of the radiation emergency plan, and the responses were investigated depending on the respondents’ characteristics (Table 1). They perceived that the national emergency medicine plan is prepared over the average level (mean = 3.55, SD = 0.74, Table 2). To examine what factors were associated with the evaluation of radiation emergency medicine, we analyzed the surveyed responses using univariate and multivariate linear regression. Female responders, inspectors, a career period of more than 16 years, and a job proportion in emergency medicine of over 80% were tightly associated with the evaluation results for national radiation emergency medicine (Table 2). Moreover, occupational roles, licensee workers, or inspectors were significantly associated with the evaluation results after controlling for sex, occupational role, career period, and job proportion in emergency medicine (B = −0.356, *p* = 0.043). None of the other variables were statistically correlated with evaluation results.

### 3.3. Different Opinions on the Radiation Emergency Medicine System between Inspectors and Licensee Workers

We compared the results of the other questions about radiation emergency medicine to explore whether occupational roles are associated with the survey response. All of the respondents assessed the radiation emergency medicine system as being at an above average level, including the safety laws, regulation policy, and emergency plan of institutes (Table 3). Interestingly, we observed some discrepancies in the surveyed responses between licensee workers and inspectors. Compared to inspectors, licensee workers favorably mentioned that the emergency medicine system for radiation accidents is well-prepared in all of the questions, especially on the governmental regulatory policy for radiation safety (*p* = 0.007 in Mann–Whitney U test). Furthermore, they also had different views on the important items of radiation emergency medicine (Figure 1). Licensee workers conceived that the workforce (34.1%) is the most important factor in the emergency medicine system, unlike inspectors, who prioritized the laws and regulations for safety (30%). Licensee workers perceived that the rescue activity (42.7%) was the highest priority for the radiation emergency medical response. Still, communication with the emergency staff (40.5%) was thought to be most critical by the inspectors.

## 4. Discussion

Here, we found that the national radiation emergency medicine plan was found to be well prepared by both inspectors and licensee workers. This consensus could have resulted from a long period of investment and efforts in radiation safety due to the social anxiety of radiation exposure because of the Fukushima NPP accidents [15,16]. Interestingly, there was some discrepancy in the evaluation of radiation emergency medicine, although it was small. Compared to inspectors, licensee workers showed a more favorable assessment in several questions about emergency medicine. An in-depth analysis demonstrated that the job proportion related to emergency medicine could be a contributing factor to the responses of licensee workers (Table 1). They had a relatively small proportion of emergency medicine as part of their job profile. Notably, while most of them (86.9%) had less than 40% of their work being relevant to radiation emergency preparedness, almost half of inspectors had over 80% of their job profile related to emergency medicine. This could imply that the facility workers’ tasks related to emergency medicine in their companies/institutes are regarded as additional minor work, which may not be considered as serious. We could not determine whether the small work proportion related to radiation emergency medicine might be detrimental to the actual operations of plans in accidents without testing in radiation emergency situations. However, some evidence found in the exercise supported that their favorable evaluation is not good enough to make it complete. Radiation emergency medical plans have been used periodically and examined by inspectors in the exercise. The inspectors always find some insufficient parts of plans and recommend that they should be revised and complemented even though licensee workers highly evaluate their own plan. Our data apparently showed that people with a small proportion of radiation emergency work favorably estimated their preparedness for radiation accidents. Furthermore, these data could suggest that licensee workers exclusively charged with emergency medicine are needed for securing a more robust system of radiation safety.

The positive perception of the current status could be evidenced by the high evaluation of the emergency medical system of the companies/institutes (question #11 in Table 3). This might be considered to be a result of complacency, as specialists believe the emergency plan is above average. Inspectors seemed to assess the emergency medical system more rigorously than licensee workers in all of the questions. They were strict in their evaluation of the government policy of safety regulation for the radiation industry even though their jobs were entrusted by the national authority, NSSC. This shows that their independent inspection activity could be guaranteed by the government, which is a basis for a fine and transparent regulatory system.

The survey results can be used to fill the perception gap between inspectors and licensee workers. As shown in Figure 1, inspectors usually examine the emergency medicine plan based on the legal regulations, which was evaluated with very low priority in the emergency response system by the licensee workers. Instead, licensee workers considered that personnel responsible for emergency medicine was highly necessary for radiation safety. In addition, inspectors perceived the communication between the emergency staff as the most important in medical radiation responses. Communication could contribute to the safety of the staff, which is a primary concern in all emergency response situations [17]. Overall, the inspectors’ views seem to focus on the efficient and sustainable response of emergency medicine at the systemic level. The different perspectives between inspectors and licensee workers stem from the actual properties of each occupational role in the regulatory system for radiation medical emergencies. To reduce this perception gap between inspectors and licensee workers in the regulatory system for radiation medical emergencies, regular technical workshops and intra- or inter-social meetings are strongly recommended to increase their understanding of differences in perception of the regulatory system, as well as to exchange information on important aspects of it. Additionally, from a regulatory point of view, governmental or institutional regulators should analyze the contributing factors of the perception of licensee workers on radiation emergency medicine and consider whether the factors should be reflected in actual laws or discipline. In addition, the Cabinet Office and Nuclear Regulation Authority are responsible for protection against nuclear disasters and enforce the various regulations at on-site and off-site locations, based on lessons learned from the 2011 Fukushima Daiichi NPP accident in Japan [18]. The importance of REM reconstruction and the information about the REM system in Japan after the Fukushima NPP accident would strengthen the importance of this study [6].

There were some limitations to our study. First, all of the inspectors were affiliated with one organization, NREMC. Considering the expertise in radiation emergency medicine, NSSC (Korean government) has entrusted NREMC with the role and responsibilities regarding the examination of the emergency plans. This could produce the perception gap in the evaluation of the licensee emergency plans for radiation accidents, even if the independent assessment is guaranteed, as mentioned above. Expansion of inspectors to other independent institutes may lead to different assessment results. Second, our survey data explained that the current level of national plans in radiation emergency medicine was estimated to be above average without a comparison of previous survey data immediately after the construction of the regulatory system. In other words, this kind of study should be performed repeatedly every two or three years to get more valuable information to produce a well-made emergency plan, regarding communication between inspectors and workers.

## 5. Conclusions

Based on our survey, we showed differences in opinions of inspectors and licensee workers in the assessment of an emergency medicine system for radiation accidents. Different views seem to come from the actual properties of each occupational role for the regulatory system. It is not easy to determine which one is more important in radiation emergency medicine. Both inspectors and licensee workers should recognize their different perceptions and exchange ideas to find a reasonable and consensual way to improve the regulatory system for radiation emergency medicine. To accomplish the accurate judgment on the emergency medical plan, a regular survey should be implemented and analyzed periodically.

## Figures and Tables

**Figure 1 ijerph-18-02458-f001:**
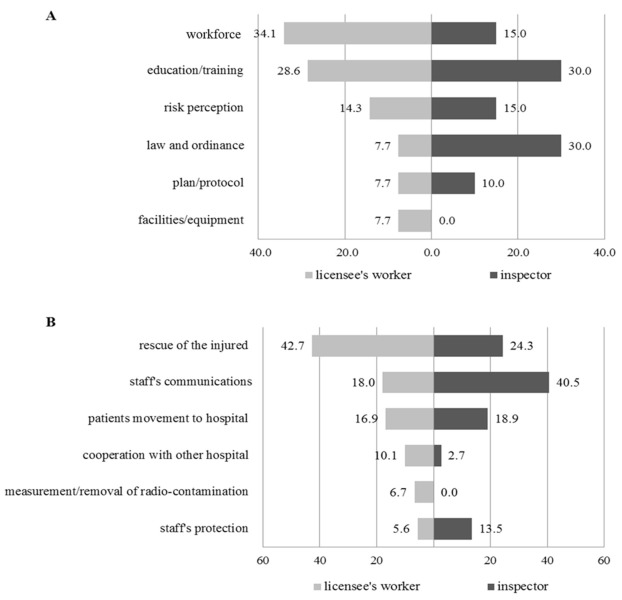
Comparison of opinions between licensee workers and inspectors. (**A**) Response to question 12, “What is the most important aspect of the regulation of emergency medicine for radiation accident?”; (**B**) response to question 13, “What is the most important factor for an effective medical response in radiation emergency?”. Workforce.

**Table 1 ijerph-18-02458-t001:** Participants’ socio-demographic characteristics.

Factor	Total		Occupational Role			
			Licensee Worker		Inspector	
	Frequency (Persons)	Percentage (%)	Frequency (Persons)	Percentage (%)	Frequency (Persons)	Percentage (%)
	133	100	92	69.2	41	30.8
**Sex**						
**Male**	100	75.2	77	83.7	23	56.1
**Female**	33	24.8	15	16.3	18	43.9
**Age (years)**						
**20~29**	9	6.8	4	4.3	5	12.2
**30~39**	61	45.9	37	40.2	24	58.5
**40~49**	35	26.3	27	29.3	8	19.5
**50~69**	28	21.1	24	26.1	4	9.8
**Education**						
**High school**	2	1.5	2	2.2	0	0.0
**Bachelor’s degree**	98	73.7	80	87.0	18	43.9
**Master’s degree**	21	15.8	9	9.8	12	29.3
**Doctorate degree**	12	9.0	1	1.1	11	26.8
**Affiliation**						
**Nuclear power generation**	58	43.6	58	63.0	0	0.0
**Nuclear fuel manufacturing**	10	7.5	10	10.9	0	0.0
**Large-scale irradiation facility**	2	1.5	2	2.2	0	0.0
**Radiation waste disposal**	1	0.8	1	1.1	0	0.0
**Research and development institutions**	62	46.6	21	22.8	41	100.0
**Period of career (years)**						
**Less than 1 year**	13	9.8	9	9.8	4	9.8
**1~5**	52	39.1	33	35.9	19	46.3
**6~10**	19	14.3	13	14.1	6	14.6
**11~15**	17	12.8	8	8.7	9	22.0
**over 16**	32	24.1	29	31.5	3	7.3
**Job proportion in medical emergencies (%)**						
**Less than 20**	77	57.9	68	73.9	9	22.0
**21~40**	16	12.0	12	13.0	4	9.8
**41~60**	10	7.5	7	7.6	3	7.3
**61-80**	10	7.5	4	4.3	6	14.6
**81~100**	20	15.0	1	1.1	19	46.3

**Table 2 ijerph-18-02458-t002:** Linear regression analyses of the factors associated with responses to the question, “Do you agree that the national plan of emergency medicine is well-prepared for nuclear or radiation accidents?”.

Factor	Number	Mean (SD) of Response	Univariate		Multivariate	
			^Ψ^ B (95% Cl)	*p*-Value	^Ψ^ B (95% Cl)	*p*-Value
**Total**	133	3.55 (0.74)				
**Sex**						
Male	100	3.65 (0.72)	Reference			
Female	33	3.24 (0.75)	−0.408 (−0.695, −0.120)	0.006 *	−0.169 (−0.538, 0.201)	0.368
**Age (years)**						
20~29	9	3.78 (0.44)	Reference			
30~39	61	3.36 (0.58)	−0.417 (−0.932, 0.098)	0.111	−0.451 (−0.968, 0.066)	0.087
40~49	35	3.63 (0.97)	−0.149 (−0.688, 0.390	0.585	−0.338 (−0.929, 0.252)	0.259
50~69	28	3.79 (0.74)	0.008 (−0.545, 0.560)	0.977	−0.398 (−1.039, 0.243)	0.221
**Education**						
High school	2	3.5 (0.71)	Reference			
Bachelor’s degree	98	3.63 (0.72)	0.133 (−0.908, 1.173)	0.801	0.611 (−0.485, 1.708)	0.272
Master’s degree	21	3.38 (0.80)	−0.119 (−1.197, 0.959)	0.827	0.502 (−0.620, 1.623)	0.377
Doctorate degree	12	3.17 (0.72)	−0.333 (−1.446, 0.779)	0.554	0.434 (−0.740, 1.608)	0.466
**Affiliation**						
Nuclear power generation	58	3.68 (0.75)	Reference			
Nuclear fuel manufacturing	10	3.60 (0.70)	−0.090 (−0.592, 0.412)	0.724	−0.09 (−0.611, 0.432)	0.734
Large-scale irradiation facility	2	3.50 (0.71)	−0.190 (−1.244, 0.864)	0.722	−0.057 (−1.090, 0.975)	0.913
Radiation waste disposal	1	4.00	0.310 (−1.168, 1.789)	0.679	0.574 (−0.863, 2.012)	0.431
Research and development institutions	62	3.40 (0.73)	−0.286 (−0.554, −0.019)	0.036	−0.42 (−0.855, 0.015)	0.059
**Occupational role**						
Licensee worker	92	3.67 (0.73)	Reference			
Inspector	41	3.27 (0.71)	−0.406 (−0.674, −0.137)	0.003 *	−0.356 (−0.702, −0.011)	0.043*
**Period of career (years)**						
Less than 1 year	13	3.23 (0.93)	Reference			
1~5	52	3.48 (0.54)	0.25 (−0.193, 0.693)	0.267	0.233 (−0.224, 0.689)	0.315
6~10	19	3.47 (0.84)	0.243 (−0.272, 0.758)	0.352	0.147 (−0.375, 0.668)	0.578
11~15	17	3.41 (0.87)	0.181 (−0.346, 0.708)	0.498	0.087 (−0.456, 0.630)	0.752
over 16	32	3.91 (0.73)	0.675 (0.205, 1.146)	0.005 *	0.461 (−0.023, 0.945)	0.062
**Job proportion in medical emergencies (percentage, %)**						
Less than 20	77	3.68 (0.75)	Reference			
21~40	16	3.38 (0.62)	−0.3 (−0.694, 0.093)	0.133	0.036 (−0.406, 0.478)	0.872
41~60	10	3.5 (0.71)	−0.175 (−0.656, 0.306)	0.472	0.045 (−0.462, 0.552)	0.861
61~80	10	2.9 (0.88)	−0.775 (−1.256, −0.294)	0.002 *	−0.345 (−0.908, 0.218)	0.227
81~100	20	3.55 (0.60)	−0.125 (−0.485, 0.234)	0.491	0.517 (−0.028, 1.062)	0.063

* *p* < 0.05. ^Ψ^ Unstandardized coefficient in the linear regression analyses. Adjusting factors: sex, occupational role, career period, and job proportion in emergency medicine. All of the items were rated using a five-point Likert scale: 1 (entirely disagree) to 5 (entirely agree). CI: confidential interval; SD: standard deviation.

**Table 3 ijerph-18-02458-t003:** Analysis of the surveyed responses for questions Q8~Q10 using Mann–Whitney U test.

Survey Question	Total			Occupational Role						Mann– Whitney U Test (*p*-Value)
				Licensee Worker			Inspector			
	N	Mean	SD	N	Mean	SD	N	Mean	SD	
Q8. Do you agree that the national plan of emergency medicine is well-prepared for nuclear or radiation accidents?	133	3.55	0.74	92	3.67	0.73	41	3.27	0.71	0.004 *
Q9. Do you think that the emergency medicine system for radiation facilities is reasonably regulated and enforced by the law?	133	3.53	0.71	92	3.59	0.74	41	3.41	0.63	0.205
Q10. Do you think that the government’s regulation policy on the radiation industry is appropriate?	132	3.44	0.67	91	3.54	0.69	41	3.22	0.57	0.007 *
Q11. Do you agree that your institute’s emergency medicine system is effective enough to protect the citizens from radiation accidents?	131	3.69	0.80	91	3.76	0.82	40	3.53	0.72	0.117

* *p* < 0.05. All of the items were rated using a five-point Likert scale: 1 (entirely disagree) to 5 (entirely agree). N: number, SD: standard deviation.

## Data Availability

The data presented in this study are available in Assessment of an Emergency Medicine System for Radiation Accidents in Korea: A State Survey of the Workers Involved the Medical Response to Radiation Accidents.

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
