# Peer review of "Assessment of an Emergency Medicine System for Radiation Accidents in Korea: A State Survey of the Workers Involved the Medical Response to Radiation Accidents"

_ijerph, 2021, doi:10.3390/ijerph18052458_

Round 1

Reviewer 1 Report

Comments to the Author

This manuscript described the survey of inspector and licensee's workers with regard to the regulatory system for radiation medical emergency.

While this is well-organized manuscript, the reviewer has the following specific comments:

Major Comments:

This manuscript demonstrates the different perspectives between inspector and licensee's workers in the regulatory system for radiation medical emergency.

Could the author suggest and discuss the concretely strategy to fill the gap between inspector and licensee's workers, and improve the situation?

Author Response

Major Comments:

This manuscript demonstrates the different perspectives between inspector and licensee's workers in the regulatory system for radiation medical emergency. Could the author suggest and discuss the concretely strategy to fill the gap between inspector and licensee's workers, and improve the situation?

=> Answer: We appreciate your excellent comments. In order to bridge the gap of the different perspectives between inspectors and licensee’s workers in the regulatory system for radiation medical emergencies, technical workshops and intra or inter-social meetings should be held regularly to increase their understanding of differences in perception of the regulatory system, as well as to exchange information on important aspects of it. Also, from a regulatory point of view, governmental or institutional regulators should analyze the contributing factors to the perception of licensee’s worker on radiation emergency medicine and consider whether the factors should be reflected in actual laws or discipline. We added the related contents in the Discussion section, in line # 234-240. Please see the revised manuscript and the details below.

“To reduce this perception gap between inspectors and licensee’s workers in the regulatory system for radiation medical emergencies, the regular technical workshops and intra or inter-social meetings are strongly recommended to increase their understanding of differences in perception of the regulatory system, as well as to exchange information on important aspects of it. Also, from a regulatory point of view, governmental or institutional regulators should analyze the contributing factors to the perception of licensee’s worker on radiation emergency medicine and consider whether the factors should be reflected in actual laws or discipline.”

Reviewer 2 Report

The study is correctly structured and the methodology is in line with the objective.

Part of the discussion includes data that could be found in the introduction, but not as a Discussion. (160-174).

In the limitations, the effectiveness of the plan of radiation medicine is discussed, which cannot be demonstrated by this study.

The conclusions do not seem relevant, although they are in line with the results obtained.

I believe that the study can be published in another journal.

Author Response

Reviewer 2: The study is correctly structured and the methodology is in line with the objective. Part of the discussion includes data that could be found in the introduction, but not as a Discussion. (160-174). In the limitations, the effectiveness of the plan of radiation medicine is discussed, which cannot be demonstrated by this study. The conclusions do not seem relevant, although they are in line with the results obtained. I believe that the study can be published in another journal.

=> Answer: Thank you for the constructive opinion. We revised the manuscript and some part of discussion (line #160-174, based on the submitted manuscript) was transferred to the appropriate location of Introduction section, in line #71-85. As you pointed out, this study does not contain the effectiveness of the plan in radiation emergency medicine but the evaluation result of the survey respondents on the questions for the radiation emergency medicine. The manuscript demonstrates the different perspectives between inspector and licensee's workers in the regulatory system for radiation medical emergency. The effectiveness of plan could be examined by table- top simulation drills of the accidents. Related part of effectiveness (line #222-225, based on the submitted manuscript) was carefully considered and revised in the manuscript. Please see the Discussion section, in line #251-#253 and the details below.  

Second, our survey data explained that the current level of national plan in radiation emergency medicine was estimated to be above the average without a comparison of previous survey data at the initial time after the construction of the regulatory system.”

Reviewer 3 Report

In my opinion this work has sufficient merits to be published in
IJERPH; I point out below only some aspects that, in my opinion, could
be improved before publication.

Abstract

In the abstract the authors refer to a global survey worldwide, however
they only refer to a particular country, Korea

Introduction

In my opinion, the introduction should be improved. Although the work is
situated in the proper context, insufficient mention is made to the
reasons that justify a survey such as the one presented by the authors.

Results

The survey is carried out to the participants in a workshop. It would be
interesting to have more data about the set of people who are dedicated
to emergency radiation medicine in Korea, showing the
ratio of those surveyed with respect to the total number of workers. An
important point is to know wether the participants in the survey are a representative sample of this set of workers.

Author Response

Reviewer 3: In my opinion this work has sufficient merits to be published in IJERPH; I point out below only some aspects that, in my opinion, could be improved before publication.

Abstract

In the abstract the authors refer to a global survey world wide, however they only refer to a particular country, Korea

=> Answer: Following you point out, we have revised the abstract more clearly to avoid confusion in abstract line 21. Please see the revised manuscript and the details below.

“The objective of this survey was to evaluate the person's perception of those who design the emergency medical plan for radiation accidents and those who supervise it in Korea.”

Introduction

In my opinion, the introduction should be improved. Although the work is situated in the proper context, insufficient mention is made to the reasons that justify a survey such as the one presented by the authors.

=> Answer: This study is thought to contribute to the construction of policy or legal regulations related to the regulatory system of radiation emergency medicine in Korea reflecting the actual requirements of licensee’s workers and regulatory agencies. Generally, this kind of survey study for the perception on the societal issues was applied to investigation on the gap-filling between interested stakeholders. We added the related content justifying the survey, together with other references in the Introduction section, in line #92-94 and 96-98. Please see the revised manuscript and the details below.

line #93-94: “This kind of survey study for the perception on the societal issues is usefully applied to investigation on the gap-filling of perspectives different the between interested stakeholders, including the public [11, 12].

Ref 11) Sjöberg, L. Risk Perception: Experts and the Public. European Psychologist. Eur Psychol 1998, 3, 1-12.

Ref 12) Tanja, P. Radiation risk perception: a discrepancy between the experts and the general population. J Environ Radioact 2014, 133, 86-91.

  •  

line #96-98: “Communication with the perception data is considered as an effective way to solve the problem of knowledge deficiency towards formulating the public’s understanding.”

Results

The survey is carried out to the participants in a workshop. It would be interesting to have more data about the set of people who are dedicated to emergency radiation medicine in Korea, showing the ratio of those surveyed with respect to the total number of workers. An important point is to know whether the participants in the survey are a representative sample of this set of workers.

=> Answer: Thank you for your comments. Total 689 persons have been enrolled as radiation emergency medical staffs in Korea (as of 19.03.31). They are affiliated to various companies and institutes, including nuclear power plant, nuclear fuel manufacturing company, large-scale irradiation facility, radio waste disposal company, R&D institute in Korea. They should legally participate in the training course for radiation emergency medicine every year for the reinforcement of the ability. The questionnaire was randomly distributed to 150 REM staffs who participated in the annual conference of REM at the KIRAMS. Some responses were collected at the on-site of the conference and the others were delivered by e-mail as a scanned file to increase the response rate for the survey. Of them, 133 staffs responded to our survey. REM staffs participated in the annual conference are representative worker entrusted by each companies and institutes with the license. We considered that their response to the survey showed the representative opinion of the REM staffs. We added the contents and revised the Results section, in line # 130-137. Please see the revised manuscript and the details below.

“REM staffs responded to our surveys should legally participate in the training course for radiation emergency medicine every year for the reinforcement of the ability. They participated in the annual conference are representative workers entrusted by each company and institute with the license. We considered that their response to the survey showed the representative opinion of the REM staffs. They are affiliated to various companies and institutes, including nuclear power plant, nuclear fuel manufacturing company, large-scale irradiation facility, radio waste disposal company, R&D institute in Korea.”

Reviewer 4 Report

05/01/2021

To the authors,

Assessment of an emergency medicine system for radiation accidents in Korea: a state survey of the workers involved the medical response to radiation accidents

Comments for the manuscript:

Your manuscript showed that there was the difference in opinions of inspectors and licensee's workers in the assessment of an emergency medicine system for radiation accidents in Korea. And these different opinions were thought to come from the actual properties of each occupational role for the regulatory system. I think that your manuscript is very unique and important. However, you should revise your manuscript in some points. Please revise your manuscript with paying attention to below comments.

Major points:

  1. (Line 113, Table 1) You showed different perspectives between inspector and licensee's workers, which stem from the actual properties of each occupational role in the regulatory system for radiation medical emergency in Korea. In addition to the values (B= -0.356, p= 0.043) of “occupational role” for licensee’s worker and inspector section in Table 2, we can also find these differences in “Job proportion of medical emergency (%)” section of Table 1. In other words, the highest frequency for licensee's worker is 77 (57.9%) for "less than 20%" and for inspector is 19 (46.3%) for "81-100%". If it's possible, you should add the information about the occupational roles (characteristics) in the section of “Results” or "Discussion".
  2. (Lines 166-174) In Japan, the Cabinet Office and Nuclear Regulation Authority are in charge of the protect for the nuclear disaster and enforce the various regulations at on-site and off-site, based on lessons learned from the 2011 Fukushima Daiichi nuclear accident. If it's possible, you should the information about the establishment of a radiation emergency medical system and/or the preparation situation for resident's evacuation in Japanese Prefectures.

| The Nuclear Regulation Authority (nsr.go.jp)

Office for the Nuclear Emergency Preparedness, Cabinet Office, Government of Japan (cao.go.jp)

Frontiers | A Simple Survey of the Preparation Situation for Resident's Evacuation in Japanese Prefectures After the Fukushima Daiichi Nuclear Power Plant Accident | Public Health (frontiersin.org)

  1. (Lines 176-178) There are very important and interesting manuscripts about the social anxiety of radiation exposure after the Fukushima Daiichi NPP accident by Japanese researchers. If it's possible, you had better add the information.

Murakami, Fukushima, anxiety radiation - Search Results - PubMed (nih.gov)

  1. Lines 216-226: You showed the survey by NREMC. I’d like to know the role of the local governments for the radiation emergency medicine systems.

Minor points:

Values

Lines 24 and 118: Is the average level mean= 3.55, SD= 0.74 or mean= 3.56, SD= 0.72 in Table 2? Which values are correct?

Keywords

Line 35: I think that it’s better to add “occupational role” and “regulatory system”.

Introduction

Line 38: After the Fukushima nuclear power plant (NPP) accident in Japan on March 2011,…

Results

Lines 113 (Table 1) and 129 (Table 2): Why is the difference number of “sex”? Male is 100 and female is 33 in Table 1. Male is 92 and female 41 in Table 2.

Line 149 (Figure 1) and Supplementary Materials: Please unify the expression of words in Figure 1 and Supplementary Materials (questionnaire), such as "man-power" and "workforce".

Discussion

Lines 199-201: What means “many people”?  the public or specialists?

Comments:

Lines 185-187: I agree with you strongly.

Author Response

Reviewer 4:

Your manuscript showed that there was the difference in opinions of inspectors and licensee's workers in the assessment of an emergency medicine system for radiation accidents in Korea. And these different opinions were thought to come from the actual properties of each occupational role for the regulatory system. I think that your manuscript is very unique and important. However, you should revise your manuscript in some points. Please revise your manuscript with paying attention to below comments.

Major points:

  1. (Line 113, Table 1) You showed different perspectives between inspector and licensee's workers, which stem from the actual properties of each occupational role in the regulatory system for radiation medical emergency in Korea. In addition to the values (B= -0.356, p= 0.043) of “occupational role” for licensee’s worker and inspector section in Table 2, we can also find these differences in “Job proportion of medical emergency (%)” section of Table 1. In other words, the highest frequency for licensee's worker is 77(57.9%) for "less than 20%" and for inspector is 19 (46.3%) for "81-100%". If it's possible, you should add the information about the occupational roles (characteristics) in the section of “Results” or "Discussion".

=> Answer: We appreciate your expertized comments and agree to your opinion. The data in Table 1 regarding to “job proportion of medical emergency” would support the conclusion that the different perspectives between inspector and licensee's workers stem from the actual properties of each occupational role in the regulatory system for radiation medical emergency. We described it in the Discussion section, in line #232-#234. In addition, our study subjects consist of workers with various occupations at different affiliation. They are radiation safety officers, administrators, medical doctor and medical researchers, including health physicists, radiation biologists, and medical specialists. As your comments, we added the occupational characteristics of respondents in the Results section, in line #137-139. Please see the revised manuscript and the details below.

Line #232-#234: “The different perspectives between inspector and licensee's workers stem from the actual properties of each occupational role in the regulatory system for radiation medical emergency.”

Line #137-#139: “Our study subjects consist of workers with various occupations at different affiliation. They are radiation safety officers, administrators, medical doctor and medical researchers, including health physicists, radiation biologists, and medical specialists.”

  1. (Lines 166-174) In Japan, the Cabinet Office and Nuclear Regulation Authority are in charge of the protect for the nuclear disaster and enforce the various regulations at on-site and off-site, based on lessons learned from the 2011 Fukushima Daiichi nuclear accident. If it's possible, you should the information about the establishment of a radiation emergency medical system and/or the preparation situation for resident's evacuation in Japanese Prefectures.

| The Nuclear Regulation Authority (nsr.go.jp) Office for the Nuclear Emergency Preparedness, Cabinet Office, Government of Japan (cao.go.jp)

Frontiers | A Simple Survey of the Preparation Situation for Resident's Evacuation in Japanese Prefectures After the Fukushima Daiichi Nuclear Power Plant Accident | Public Health (frontiersin.org) + https://doi.org/10.3389/fpubh.2020.496716

=> Answer: We agree with your opinion. The information about the REM system of Japan experienced in Fukushima Daiichi nuclear accidents would strengthen the importance of this study. As your comments, we added the references to Introduction section with a relevant context. Please see the main body of Discussion section, in line #240-#244 and the details below.

“In addition, the Cabinet Office and Nuclear Regulation Authority are responsible for protection against nuclear disaster and enforce the various regulations at on-site and off-site, based on lessons learned from the 2011 Fukushima Daiichi NPP accident in Japan [18]. The importance of REM reconstruction and the information about the REM system in Japan after the Fukushima NPP accident would strengthen the importance of this study [6].”

Ref 18) Tsujiguchi, T.; Sakamoto, M.; Koiwa, T.; Suzuki, Y.; Ogura, K.; Ito, K.; Yamanouchi, K.; Kashiwakura, I. A. Simple Survey of the Preparation Situation for Resident's Evacuation in Japanese Prefectures After the Fukushima Daiichi Nuclear Power Plant Accident. Front Public Health 2020, 2, 8, 496716.

  1. (Lines 176-178) There are very important and interesting manuscripts about the social anxiety of radiation exposure after the Fukushima Daiichi NPP accident by Japanese researchers. If it's possible, you had better add the information. Murakami, Fukushima, anxiety radiation - Search Results -PubMed (nih.gov)

=> Answer: We think that your recommended references would help the understanding of the importance of this kind of survey study. Some part of the Introduction section was revised and added new relevant references to complement the contents. Please see the text in the Introduction section line #94 -#96 and the details below.

“Previous research about the social anxiety and poor quality of life after the NPP accident by Japanese researchers reflect concerns about exposure to radiation [13, 14].

Ref 13) Takebayashi, Y.; Lyamzina, Y.; Suzuki, Y.; Murakami, M. Risk Perception and Anxiety Regarding Radiation after the 2011 Fukushima Nuclear Power Plant Accident: A Systematic Qualitative Review. Int J Environ Res Public Health. 2017, 14, 1306.

Ref 14) Murakami, M.; Takebayashi, Y.; Takeda, Y.; Sato, A.; Igarashi, Y.; Sano, K.; Yasutaka, T.; Naito, W.; Hirota, S.; Goto, A.; Ohira, T.; Yasumura, S.; Tanigawa, K. Effect of Radiological Countermeasures on Subjective Well-Being and Radiation Anxiety after the 2011 Disaster: The Fukushima Health Management Survey. Int J Environ Res Public Health. 2018, 15, 124.

  1. (Lines 216-226): You showed the survey by NREMC. I’d like to know the role of the local governments for the radiation emergency medicine systems.

=> Answer: The role of local governments in Korea for the radiation emergency medicine system is to implement the protective actions based on providing assistance for the medical system, including the support of emergency service systems for patients transfer, contamination screening and decontamination of residents before and after moving into shelter, psychological counseling, and potassium iodine distribution and monitoring of side effects. We briefly added the related contents in the Introduction section, in line # 51-55. Please see the revised manuscript and the details below.

“The role of local governments in Korea for the radiation emergency medicine system is to implement the protective actions based on providing assistance for the medical system, including the support of emergency service systems for patients transfer, contamination screening and decontamination of residents before and after moving into shelter, psychological counseling, and potassium iodine distribution and monitoring of side effects.

Minor points:

Values

Lines 24 and 118: Is the average level mean= 3.55, SD= 0.74 or mean= 3.56, SD= 0.72 in Table 2? Which values are correct?

=> Answer: We are sorry to make confusion. The average level mean= 3.55, SD= 0.74, and we corrected the value in Table 2.

Keywords

Line 35: I think that it’s better to add “occupational role” and “regulatory system”.

=> Answer: As your comments, we added the “occupational role” and “regulatory system” as keywords.

Introduction

Line 38: After the Fukushima nuclear power plant (NPP) accident in Japan on March 2011,…

=> Answer: As your comments, we mentioned specific times in the sentence.

Results

Lines 113 (Table 1) and 129 (Table 2): Why is the difference number of “sex”? Male is 100 and female is 33 in Table 1. Male is 92 and female 41 in Table 2.

=> Answer: Sorry to make this error. It was typing error. We corrected the value in Table 2.

Line 149 (Figure 1) and Supplementary Materials: Please unify the expression of words in Figure 1 and Supplementary Materials (questionnaire), such as "man-power" and "workforce".

=> Answer: We revised Figure 1 by workforce to unify the expression of words.

Discussion

Lines 199-201: What means “many people”? the public or specialists?

=> Answer: We changed the “many people” to “specialists” following your comments.

Comments:

Lines 185-187: I agree with you strongly.

Þ Yes. This point of discussion should be further investigated continually in next year.

We deeply appreciate your review of this work. We are hopeful that this revision will satisfy the criteria for publication of the Editorial board of the journal.

Round 2

Reviewer 2 Report

The study is correctly structured and the methodology is in line with the objective.

In the limitations, the effectiveness of the plan of radiation medicine is discussed, which cannot be demonstrated by this study.

The conclusions do not seem relevant, although they are in line with the results obtained.

I believe that the study can be published in another journal

Author Response

Comments and Suggestions for Authors

The study is correctly structured and the methodology is in line with the objective.

In the limitations, the effectiveness of the plan of radiation medicine is discussed, which cannot be demonstrated by this study.

The conclusions do not seem relevant, although they are in line with the results obtained.

I believe that the study can be published in another journal

=> Answer: Thank you for your opinion. As your comments, this study does not contain the effectiveness of the plan in radiation emergency medicine but the evaluation result of the survey respondents on the questions for the radiation emergency medicine. The effectiveness of plan could be examined by table-top simulation drills of the accidents, which is thought to be beyond of this study. This information could contribute to the well-operation of regulatory system for the plan of radiation emergency medicine. Please consider the manuscript positively in the view of objective.

Reviewer 4 Report

To the authors,

I have no more specific comments. However, I found the minor point in numbering in Line 168 (3.3 ?).

Author Response

To the authors,

I have no more specific comments. However, I found the minor point in numbering in Line 168 (3.3 ?).

=> Answer: We have renumbered “3.2” to“ 3.3”, following you point out.
